# Deep Learning to Classify AL versus ATTR Cardiac Amyloidosis MR Images

**DOI:** 10.3390/biomedicines11010193

**Published:** 2023-01-12

**Authors:** Philippe Germain, Armine Vardazaryan, Aissam Labani, Nicolas Padoy, Catherine Roy, Soraya El Ghannudi

**Affiliations:** 1Department of Radiology, Nouvel Hopital Civil, University Hospital, 67091 Strasbourg, France; 2ICube, University of Strasbourg, CNRS, 67000 Strasbourg, France; 3IHU, 67000 Strasbourg, France; 4Department of Nuclear Medicine, Nouvel Hopital Civil, University Hospital, 67091 Strasbourg, France

**Keywords:** cardiac amyloidosis, light chain, transthyretine, deep learning, convolutional neural network, algorithm vs. human comparison

## Abstract

The aim of this work was to compare the classification of cardiac MR-images of AL versus ATTR amyloidosis by neural networks and by experienced human readers. Cine-MR images and late gadolinium enhancement (LGE) images of 120 patients were studied (70 AL and 50 TTR). A VGG16 convolutional neural network (CNN) was trained with a 5-fold cross validation process, taking care to strictly distribute images of a given patient in either the training group or the test group. The analysis was performed at the patient level by averaging the predictions obtained for each image. The classification accuracy obtained between AL and ATTR amyloidosis was 0.750 for cine-CNN, 0.611 for Gado-CNN and between 0.617 and 0.675 for human readers. The corresponding AUC of the ROC curve was 0.839 for cine-CNN, 0.679 for gado-CNN (*p* < 0.004 vs. cine) and 0.714 for the best human reader (*p* < 0.007 vs. cine). Logistic regression with cine-CNN and gado-CNN, as well as analysis focused on the specific orientation plane, did not change the overall results. We conclude that cine-CNN leads to significantly better discrimination between AL and ATTR amyloidosis as compared to gado-CNN or human readers, but with lower performance than reported in studies where visual diagnosis is easy, and is currently suboptimal for clinical practice.

## 1. Introduction

Cardiac amyloidosis (CA) is related to the extracellular infiltration of insoluble misfolded proteins, the accumulation of which alters the structure and function of the heart, leading to restrictive cardiomyopathy and heart failure. Amyloidosis is a systemic disease also involving other organs such as the nervous system, musculo-skeletal soft tissues and kidneys in particular [1]. The type of proteins involved are variable, but in more than 95% of cases, they are monoclonal immunoglobulins light-chains (AL amyloidosis) produced by plasma cells or transthyretin (plasma transport protein) produced by the liver (ATTR amyloidosis) [2,3,4]. This serious disease is life-threatening, with an average survival ranging from 6 to 94 months in light-chain amyloidosis (median survival > 5 years), according to staging score [3]. ATTR CA is slowly progressive and has better prognosis than AL CA [3].

Accurate identification of the amyloid subtype is critical for determining the appropriate treatment, which is specific for AL or for ATTR. Reference methods proceed from histological examination with immunohistochemical, immunofluorescence, immunoelectron microscopic techniques and, more recently, mass spectrometry [5]. Light-chain AL CA requires prompt chemotherapy (±followed by autologous stem cell transplantation /anti-plasma cell or anti-amyloid antibodies), providing a marked improvement in survival. For the ATTR subtype, two kinds of medications may reduce tissular transthyretin deposition, thus decreasing disease progression, morbidity and mortality. One way consists of the degradation of RNA transthyretin hepatic synthesis (Patisiran, Inotersen) [3,4]. The other way is to stabilize the tetrameric form of transthyretin, thus preventing its dissociation and misfolding (Tafamidis) [6].

The diagnosis of cardiac amyloidosis is based on algorithms proposed by the American [7,8] and European societies of cardiology [4], in which echocardiography and cardiac MRI play important roles, along with blood analysis, in seeking for monoclonal gammopathy and bone scintigraphy. In particular, the diagnostic process has gained important improvements thanks to the recognition of ^99m^Tc pyrophosphate or diphosphonate SPECT imaging for the identification of the ATTR form of the disease [8,9]. This is why this disease, which was thought to be rare, is more and more often identified and is now recognized as a main cause of heart failure in elderly patients with preserved ejection fraction [10].

Cardiac MRI is an important tool in the diagnosis of cardiac amyloidosis, thanks to the rich semiology provided. Alongside fairly characteristic but non-specific structural elements (mainly left ventricular wall thickening), the cornerstones of MRI diagnosis are the typical pattern of myocardial late-enhancement after gadolinium chelates injection (LGE) [11,12] and marked increase in extracellular volume (ECV) [13,14]. However, MR-based distinction between AL and ATTR subtypes is unreliable in practice [15] and new differential features in this direction would be helpful.

Deep learning has shown interesting potential in many fields of medical imaging. Its diagnostic capacities, similar or even superior to those of experienced practitioners, have been demonstrated in multiple fields, for example malignancy risk estimation of pulmonary nodules [16], low grade vs high grade glioma differentiation [17], melanoma image classification [18], benign vs malignant renal tumor discrimination [19] or breast cancer diagnosis [20].

Up to now, six studies were devoted to deep learning distinction between cardiac amyloidosis and other pathological conditions, based on echocardiography [21], MRI [22,23,24] and on nuclear imaging [25,26]. Reported AUC of the ROC curve was almost always >90%, depending on the method and the studied population. However, there is little data on the differential diagnosis between AL amyloidosis and ATTR amyloidosis and above all, there is no comparison between CNN classification vs human diagnosis in this area. This is why we undertook a comparative study of the performance of a classical popular CNN compared to the performance of experienced practitioners to identify these two forms of cardiac amyloidosis from MR images.

## 2. Materials and Methods

### 2.1. Study Population

From the clinical reports available in our hospital, patients with proven cardiac amyloidosis between January 2010 and June 2022 were selected and the corresponding Dicom images were extracted from the PACS. This retrospective study was registered and approved by the Institutional Review Board of our university hospital and all datasets were obtained and de-identified, with waived consent in compliance with the rules of our institution.

The diagnosis of cardiac amyloidosis was established in accordance with European [4] and American recommendations [7,8]. One hundred and twenty patients were selected for this study, including seventy cases of AL amyloidosis and fifty cases of ATTR amyloidosis. All patients had symptoms of heart failure and presented with one or several characteristic echocardiographic features, including thickened myocardial walls, enlarged atria, sparkling aspect of the septum, target-like pattern on longitudinal strain map with apical sparing and more or less restrictive profile at doppler examination.

CMR features were also suggestive or typical for all included patients (some cases without awaited MR findings despite clinical, biological or echocardiographic suggestive data were excluded from the study). Most important MR features were thickened septum, generally associated with left ventricular mass increase (except for 5 patients with AL CA whose septal thickness was <12 mm, which is concordant with previously reported observation [27]), left/right atrial dilatation, reduced downward systolic excursion of the mitral plane, pericardial or pleural effusion, increased native T1 myocardial relaxation time and extracellular volume (ECV) and typical late gadolinium enhancement of the ventricular ± atrial cardiac walls with inversed myocardial vs blood pool nulling in TI scout Lock–Locker sequences.

AL amyloidosis diagnosis was based on the presence of a monoclonal dyscrasia (blood and urine immunofixation, kappa and lambda immunoglobulin free chains quantitation), in patients with heart failure symptoms and suggestive cardiac imaging features. Amyloidosis deposits were proven in biopsy samples of the gingival tissue, salivary gland, subcutuaneous abdominal fat, kidney, rectal mucosa and heart.

ATTR amyloidosis was assessed in patients without monoclonal gammopathy, with ^99m^Tc-diphosphonate bone SPECT Perugini score >1 and/or with amyloid deposits on extracardiac and/or endomyocardial biopsy (3 cases, including 1 transplant and 1 necropsy). Histological diagnosis was available in 4 patients with concomitant MGUS. 

Patients’ characteristics are listed in Table 1. As previously reported, the disease was predominant in males (especially for ATTR subtype, 94%). Patients with ATTR CA presented older, with almost 9 kg weight greater, markedly increased septal thickness and left ventricular mass index and increased ECV as compared to AL patients. Pericardial effusion (×2), pleural effusions (×2.5) and pleural + pericardial effusions (×3.8) were clearly predominant in case of AL CA.

### 2.2. MR Image Acquisition and Dicom Image Preparation

Standard cine-CMR acquisitions were performed using 1.5 Tesla scanners (3 Siemens and 1 Philips systems). Cine MR sequences were obtained with steady state free precession sequences (TE/TR 1.6/3.5 ms). Eight to thirty-two elements cardiac coil were used. Vertical long axis (VLA), 4-chamber view (4C) and a stack of short axis views (SA) were acquired with 6 to 8 mm thick slices. 

Gadolinium-enhanced sequences were obtained in 113 patients only, owing to renal insufficiency precluding gadolinium injection in 7. A short axis TI-scout Look–Locker sequence was used to determine if myocardial nulling occurs before or after blood pool nulling and helped to adjust the inversion time for further inversion-recovery late gadolinium images (LGE). As this was often tricky in CA patients, TI = 200 ms was often chosen. If PSIR sequences were used, only magnitude component (not PSIR image itself) was considered for this study.

A dedicated home-build software (Visual-C) was used to read Dicom files, de-identify patient’s name and birth date, rescale all images as to set pixel size to a homogeneous 1.5 mm size, crop the centered cardiac region to 160 × 160 pixels dimensions and visually set the gray level (windowing). End diastole and end systole frames from the cine sequences were visually selected and stored (2 frames per cine sequence). For LGE sequences, 1 to 10 frames were selected according to technical quality and to left ventricular coverage.

Thus, 1436 frames included in 718 cine-MR sequences were processed for the 120 patients. For LGE sequences, 2564 frames included in 540 directories from 113 patients were processed. The preparation software also allowed for the assignment of the labels intended for subsequent classification by deep learning: orientation and AL or ATTR CA subtypes.

### 2.3. Deep Learning Process

Python 3.7.6, with Keras library and TensorFlow backend were used for CNN implementation. Attention was paid as to strictly distribute images of a given patient in either the train set or the validation set in concordance with CLAIM recommendations (item 21) [28]. VGG16 base model [29] was used, followed by layers: Dense 512, Dropout 0.50, Dense 256, Dropout 0.50, Dense 64, Dropout 0.50, Dense 1 and finally output Sigmoid activation layer.

Hyperparameters (200 Epochs, parameters of the image data generator, SGD optimizer, batch size 32, drop-out rate, learning rate 6 × 10^−5^, decay 10^−6^, number of trainable layers) were identical to those previously chosen on similar data sets [24]. Binary cross entropy was used as a loss function. The parameters of data augmentation applied during training were zoom range < 0.15, 15% height and width shift range and up to 20° rotation.

Classical 5-fold cross validation was used for both cine and LGE images (first green circle in Figure 1). Accuracy and loss were the two metrics calculated for the frame-based analysis. From each model trained through the 5-fold cross validation process, the predicted probability of each frame in the corresponding test set was calculated.

### 2.4. Parsing Flowchart

In order to approach the way in which the diagnosis is carried out by human operators, the analysis of the results by frames was completed by two subsequent steps. The prediction per patient was calculated by averaging the predictions per frame of each patient (green circle in the middle of Figure 1). Finally, given that human diagnosis proceeds from the combination of cine and LGE images, the calculation of the combined prediction per patient was carried out, (1) by simple averaging and (2) by logistic regression of the cine and LGE predictions (green circle in the top right of Figure 1). 

Cine and LGE panels containing all frames of a given patient (cine and LGE) were also prepared and randomly shuffled in order to be scored by human observers. Panels blind reading were performed by one radiologist and by two cardiologists (>10 years’ experience of CMR analysis and reporting).

In order to localize the main pixel areas from which CNN classification emerges, saliency class activation maps were computed according to the GradCam algorithm [30]. This method consists of first, creating a model that maps the input image to the activations of the last convolutional layer; second, create a model that maps the activations of the last convolutional layer to the final class predictions; and finally, computing the gradient of the top predicted class for the input image with respect to the activations of the last convolutional layer. 

Thus, is it possible to identify the most contributing anatomical regions that led to the correct or erroneous classification made by the CNN. Saliency map computations were only performed for cine images.

### 2.5. Statistical Analysis

Computed performance metrics were accuracy, confusion matrix and Receiver Operating Curves (ROC), with corresponding area under the curve (AUC) values. Categorical data were compared through Chi-square test. Continuous variables were compared by Student’s t test. Comparisons of AUC values were carried out with Delong test. Statistical analyses were performed using MedCalc 12.1.4 (MedCalc Software, Ostend, Belgium).

## 3. Results

### 3.1. Classification of Cine-MR Images

Image classification based on a common model including all orientation planes of patient-based cine-MR is illustrated in Figure 2. This model provided an accuracy of 0.750 and AUC of 0.839 (Table 2). Weaker results were obtained if specific models were trained only on the long axis images (acc 0.630, AUC 0.722) or on the short axis orientation views (acc 0.686, AUC 0.797).

### 3.2. Classification of LGE Images

Overall classification of patient-based LGE images, illustrated in Figure 3, provided a lower accuracy of 0.611 and AUC of 0.679 (*p* = 0.0041 as compared with cine data).

### 3.3. Classification of Combined Cine and LGE Images

Per-patient averaging of the prediction probabilities obtained with the cine model and with the LGE model provided: Acc 0.708, AUC 0.821. A similar result was obtained with logistic regression: Acc 0.742, AUC 0.818 (Table 2).

### 3.4. Blind Reading by Experienced Radiologist/Cardiologist

Comparison between amyloidosis subtypes classification obtained by experienced radiologist/cardiologists and by the best CNN (cine model) is given in Table 3. 

CNN provided significantly better performance as compared to human readers. Considering the best human reader and the best CNN model led to accuracies of 0.675 vs. 0.750 and AUCs of 0.714 vs. 0.839 (*p* = 0.0075). Corresponding ROC curves of these comparisons are plotted in Figure 4.

### 3.5. Analysis of Saliency Maps for cineMR Images

Saliency maps, which reveal the pixel areas responsible for classification, illustrated in Figure 5, show that cardiac regions contribute to CNN decisions in 68% of cases correctly classified and in 60% of cases erroneously classified (Chi-square 5.18, *p* = 0.023). Other anatomical regions mainly targeted by the CNN were abdominal or chest wall fat (approx. 17%), lung (approx. 9%), liver (approx. 8%) and stomach (approx. 2%), with distribution being quite similar for correct classification (concordant) and erroneous classification (discordant).

## 4. Discussion

The aim of this work was to test the hypothesis that convolutional neural networks, capable of seeing features imperceptible to the naked eye, could classify the two types of cardiac amyloidosis (AL and ATTR) better than an experienced human observer, which is important because the treatments of these two conditions are completely different. Our results indeed show a capacity of CNNs superior to that of human observers but lower than those reported in studies where visual diagnosis is easy (e.g., 22, 23, 25), not yet allowing its application for clinical diagnosis.

### 4.1. Differential Diagnosis between AL and ATTR Cardiac Amyloidosis

The diagnosis of cardiac amyloidosis by MRI could be easy in advanced forms of the disease thanks to the richness and multiplicity of semiological indices. The performance reported in the literature in comparison with other hypertrophic heart disease is excellent (high sensitivity of 95% and an even higher specificity of 98% [11]); thus, MRI is one of the preferred tools in the diagnostic algorithms of the ESC [4]. Conversely, the differential diagnosis between AL and ATTR cardiac amyloidosis is not reliable on MRI [15]. In practice, this diagnosis is mainly based on the search for a monoclonal dyscrasia, which points to AL forms, and on bone scintigraphy, which identifies ATTR forms [9]. Positon emission tomography (PET) with ^18^F Florbetaben or Florbetapir seems to be complementary with bone scintigraphy by preferentially identifying AL forms of cardiac amyloidosis [8,26].

Several works have sought to point out some MR features allowing the two forms of CA to be distinguished. Morphologically, septal thickening is more increased in case of ATTR amyloidosis, resulting from known increased amyloid burden in this subtype [31,32,33], as also shown by our data. The pattern of myocardial LGE would be more diffuse and extensive in the AL subtypes [33] and right ventricular ejection fraction combined with age appeared also quite discriminative [34]. The prevalence of pericardial and pleural effusion is also greater in AL forms [35], consistent with our data. 

Finally, the increase in the T2 relaxation time of the myocardium, in connection with myocardial edema linked to the direct cardiotoxic effect of amyloidogenic light chains [3,36], has also been reported [37], but does not appear to be very discriminating in our series. Faced with these limitations, it would be desirable to have better tools to ensure the differential diagnosis between the subtypes of cardiac amyloidosis.

### 4.2. The CNN Tool Seemed to Be Promising in This Area

Surprising results have been reported in the literature, with CNNs able to identify clues not discernible to the naked eye, thus surpassing human experts in disease diagnosis. For example, in differentiating benign from malignant renal tumors, based on T2-weighted images, higher AUC was obtained with the CNN model (0.906) as compared with the AUC obtained by 2 radiologists (0.724) [19]. Similarly, an effective histological approach, unaffordable to the simple visual reading of MRI images, has been demonstrated to establish the Furhman grading of renal cancers [38] and to establish the Gleason score of prostate cancers [39]. CNN also demonstrated higher AUC than the usual Toronto score for hypertrophic cardiomyopathy mutation prediction using cardiac cine-MR images [40].

### 4.3. CNN Results for the Diagnosis of Cardiac Amyloidosis

To date, six studies devoted to the diagnosis of cardiac amyloidosis by CNN have been published. Analysis of four-chamber echocardiographic views yielded an AUC > 0.9, surpassing human diagnosis [21]. By MRI, the work of Martini et al. [22] showed an AUC of 0.98 comparing LGE of 106 patients with amyloidosis vs. 100 patients with other hypertrophic heart diseases using CNN models specific to long-axis or short axis orientation. Agibetov et al. obtained similar results in 82 amyloidosis vs 420 patients with various other heart diseases, with little difference between cine, LGE and T1 mapping images [23]. The classification by CNN and by expert radiologists of cine-MR images of patients with CA vs other causes of LVH showed the clear superiority of CNNs (AUC 0.825 vs. 0.727) [24]. Good performance of CNNs has also been reported for CA with bone scintigraphy and with ^18^F Florbetaben [25,26]. So far, however, no study by CNN has been devoted to the classification of CA subtypes.

For CA subtypes classification, the present work shows an AUC of 0.839 from cine images but only 0.679, similar to human results, considering LGE images. The results obtained by the combination (simple mean or logistic regression) of cine and LGE data do not improve the results. 

Several explanations can account for the poor results obtained with LGE images. First of all, it is important not to compare the poor performance obtained in this study by LGE (AUC 0.679) with the excellent results reported by Martini et al. [22] or Agibetov et al. [23] (AUC > 0.96). In these two studies, LGE images were radically different between the CA group (diffuse or subendocardial hyperenhancement) and the control group (no myocardial hyperenhancement), while in our study, a similar myocardial hyperintensity was present in both groups of patients.

Moreover, the difficulties of obtaining good quality images are well known in the case of CA. The marked reduction in the T1 relation time in the myocardium may exceed that in the blood pool and the setting of the TI inversion time is tricky. This is why Martini et al. used a specific preparation that optimized the quality and stability of LGE images, which was not done in our protocol. Thus, a fairly significant variability was observed in the quality of our LGE images, which may have mitigated any specific differences between the AL forms and the ATTR forms. Be that as it may, the visual analysis by human observers was hardly guided by the appearance of the LGE, but rather by the morphological aspects observed on Cine images.

### 4.4. Possible Explanations for CNN Distinction between AL and ATTR Cardiac Amyloidosis

At a macroscopic scale, structural differences are known between the two types of CA: less myocardial thickening is observed and LGE is more extensive in AL than in ATTR subtypes. These features helped the human observer to discriminate—albeit with low accuracy—between these two disease states and, similarly, CNN could extract corresponding relevant differential features. Moreover, pericardial or pleural effusion could account for additional discriminative features (but saliency maps did not target specifically on effusion area).

At a molecular scale, misfolded proteins are initially generated and aggregated as protofibrils and ultimately mature fibrils, containing several common compounds, leading to persistent amyloid deposits and explaining their affinity for Congo red and their yellow-green birefringence under polarized light [36]. However, pathobiology of deposition is different in AL and in ATTR amyloidosis. Larsen et al. reported histopathological differences due to distinct deposition patterns according to amyloid type. Thus, as compared with ATTR subtype, AL deposits mostly presented as broad zones of “uniform lace-like deposition around individual cardiomyocytes”, diffusely infiltrating peri-myocyte connective tissue with a “chicken-wire” pattern, whereas they appear more nodular, patchy, interspersed by uninvolved myocardium in ATTR subtype [5]. Moreover, in AL forms, these deposits are more severe and more located in arterial and venous locations, as well as in endocardial regions, as compared with ATTR CA [5]. However, the author pointed out that a mixture of patterns of interstitial deposition commonly occurs, limiting the clinical applicability of this approach.

Such tissular distribution may perhaps explain that deep learning extracted features allow to discriminate both pathological conditions. However, this assumption is uncertain due to the scale of analysis since histopathologic differences pointed out by Larsen et al. is at a sub-millimetric level, whereas MR images resolution is typically 2 mm in plane. Another cause for doubt about this hypothesis lies in the targeting of salient regions by the GradCam algorithm, which often correspond to extracardiac structures or to the blood pool, rather than the myocardium itself (e.g., Figure 5). However, being a systemic disease, amyloidosis also affects other structures, such as subcutaneous fat or lungs. Spleen or liver involvement was also reported in 41% of CA (mostly in the AL forms) [41].

### 4.5. The “Black-Box” Nature of CNNs Remains a Major Concern

Owing to the multilayer nonlinear structure and high complexity level of the great number of weights involved in CNN models (14,714,688 parameters in the VGG model), it remains presently impossible to discern the nature and interplay of relevant CNN parameters. This remains a definitely redhibitory limitation inherent to the “black-box” nature of CNN models. Thus, as with all studies concerning the classification of pathologies by CNNs, we remain frustrated by the lack of explainability of the precise reasons for the network decision. This is a general but persistent problem, without much conceptual progress in the last years, limiting the acceptability of these techniques in the medical field, which requires understandable arguments to be credible.

The return to better explainable predictive models such as decision trees has been proposed [42], but is not applicable to imaging data. Multiple methods have been proposed to improve the explainability of CNNs, for example, pixel-attribution-maps, which can proceed in a forward way (e.g., Local Interpretable Model-agnostic Explanations or LIME) or in a backward way (gradient, e.g., GradCam saliency maps, such as used here). Adding a probabilistic layer to a model’s architecture may also help evaluate the uncertainty of this model’s predictions.

Radiomics, proceeding from precisely known features handcrafted in advance, can constitute a positive approach in this field. Having a limited number of filters, they oppose CNNs, which are capable of picking up countless non-human-interpretable features. Conversely, a disadvantage of radiomics is the need to guide the process on myocardial regions of interest, which constitutes an additional preprocessing step (but which could benefit from automatic segmentation by U-net). However, radiomics offers the advantage of better explainability and has proved able to efficiently discriminate between hypertensive heart disease and hypertrophic cardiomyopathy [43] or between recent infarction vs. old infarction [44].

### 4.6. Study Limitations

This work should, in no way, be considered as an alternative to the well-established diagnostic algorithm based on biological data and bone scintigraphy for cardiac amyloidosis. The aim of the work was only to position deep learning in relation to human visual abilities to discriminate between the two main varieties of amyloidosis.

The detailed analysis of the probability scores obtained for each image shows a marked skewness in the histogram distribution, with values concentrated around 0 (AL) and around 1 (ATTR). This is consistent for the binary classification. However, classification instabilities are often observed, with prediction probability scores that can go from one extreme to the other, as illustrated by the examples in Figure 2, where we can see a large difference in scores, from 0.953 to 0.088, between images 8 and 12, even though few differences appear visually between these two images. This instability, which is not corrected by adjusting the parameters of the fully connected layers, probably reflects an insufficient number of observations in the data set. Other CNN designs should be tested and the number of observations should be increased.

The very different prevalence of pericardial and pleural effusions (2 and 2.5 times more frequent in the AL forms, slightly higher than in the study of Binder et al. [35]) could be considered as a confounding factor, but rather constitutes a diagnostic signature that can be analyzed in a similar way by the CNN and by humans.

The monocentric nature of this study constitutes another limitation. Multicenter studies would be preferable and, moreover, external validation is mandatory for the generalization of our findings.

### 4.7. Perspectives

Like many other studies in this field, this work constitutes a small step for the improvement of diagnosis, concentrated here on the very specific subject of the distinction between the two main forms of cardiac amyloidosis from MRI data. A fairly basic method exploiting classical transfer learning with the VGG algorithm was used on a quite small number of observations. This results in a significant improvement in classification compared to human diagnosis, which, once again, confirms the remarkable capabilities of CNNs.

It would be surprising if these capabilities of CNNs, superior to those of humans, did not gradually lead to the implementation of these techniques in future generations of scanners and MRIs.

However, there will be some hurdles to overcome. First, it is essential that such studies be more consistent, solid, based on a greater number of observations from several centers in order to improve the generalizability and the credibility of the results. External validation is also essential [28]. These requirements, however, constitute a significant gap between the category of small exploratory clinical studies—of which the present work falls—and generalizable robust works intended to be integrated into future imaging and diagnostic devices. These challenges require that they be conducted within the framework of large-scale institutional or industrial projects. It will happen, but slowly.

Along with building databases and models that surpass human capabilities, neural network theorists will also need to find better methods that shed light on the explicability of automatic classifications. The credibility of these methods, and therefore their practical use, depends on it. This is all the more important since the screening by algorithm and big data is in contradiction with the scientific paradigm that has always prevailed until now, namely: hypothesis → experimentation → conclusion, as explained by Claude Bernard in his “Introduction to the Study of Experimental Medicine” (1865).

Finally, when a notification suggesting a probability of diagnosis such as “type AL cardiac amyloidosis” appears on the scanner screen during the course of the examination—with as much intelligible and verifiable arguments as possible—the clinician should always keep a critical mind and take responsibility for the final diagnosis.

## 5. Conclusions

In this study, we showed the superiority of convolutional neural networks compared to visual reading by experienced human operators, to classify the two main forms of cardiac amyloidosis, AL and ATTR, from MR images (mainly cine rather than LGE). 

For this classification, the algorithm outclasses humans by more than 10 points (in terms of accuracy and AUC), but the maximum AUC obtained with the CNN (0.839) remains below the performance generally reported in many other diagnostic fields. This work constitutes a small additional step in the great quest for complementary means of diagnosis in radiology and reinforces the hopes brought about by neural networks, even if significant practical and theoretical obstacles (explainability) still hinder their widespread implementation in medical devices.

## Figures and Tables

**Figure 1 biomedicines-11-00193-f001:**
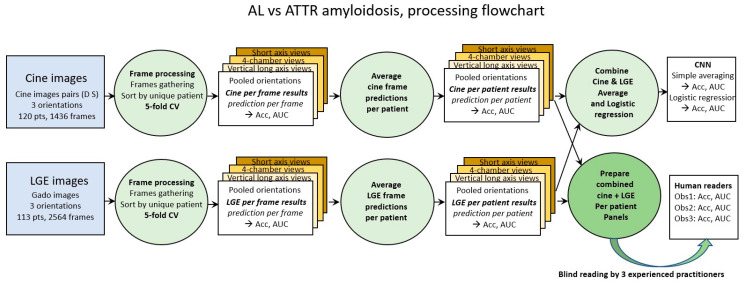
Processing steps flowchart. Cine images and LGE images are first processed per frame (green circles on the left). Averaging prediction of all frames of a given patient provides prediction per patient (green circle in the middle). Combination of cine and LGE prediction is performed by (1) simple averaging and (2) logistic regression (upper right green circle). Panels of cine frames and of LGE frames of each patient are also prepared and shuffled to be blindly read by the 3 human observers (lower right green circle).

**Figure 2 biomedicines-11-00193-f002:**
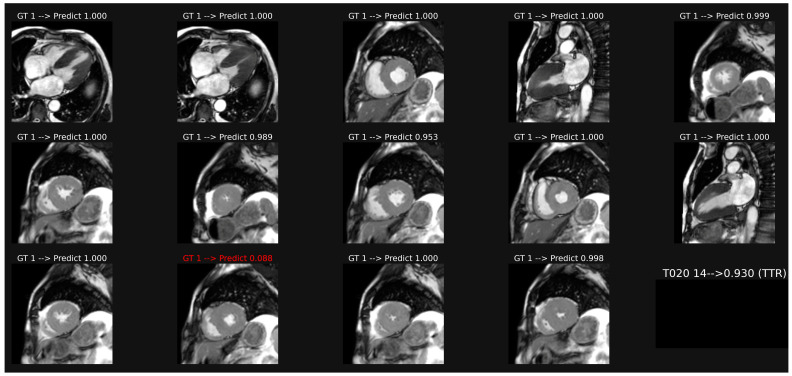
Example of classification of cine images in a patient with ATTR amyloidosis. The average of the probabilities over all the images leads to a score (0.93 here), which corresponds to a correct classification in this case, despite one misclassification in image 12.

**Figure 3 biomedicines-11-00193-f003:**
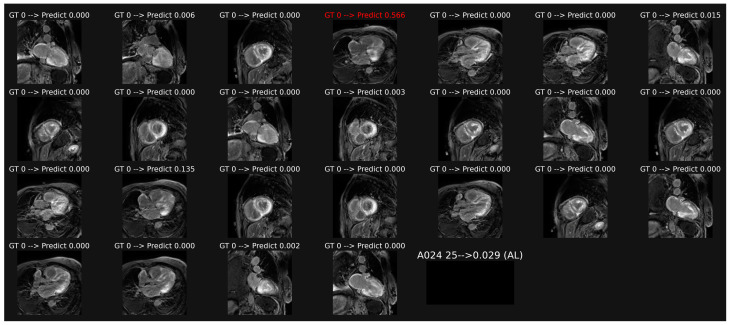
Example of classification of LGE images in a patient with AL amyloidosis. The averaging of the probabilities over all the images leads to a score (0.029 here) which corresponds to a correct overall classification in this patient, despite one misclassification in image 4.

**Figure 4 biomedicines-11-00193-f004:**
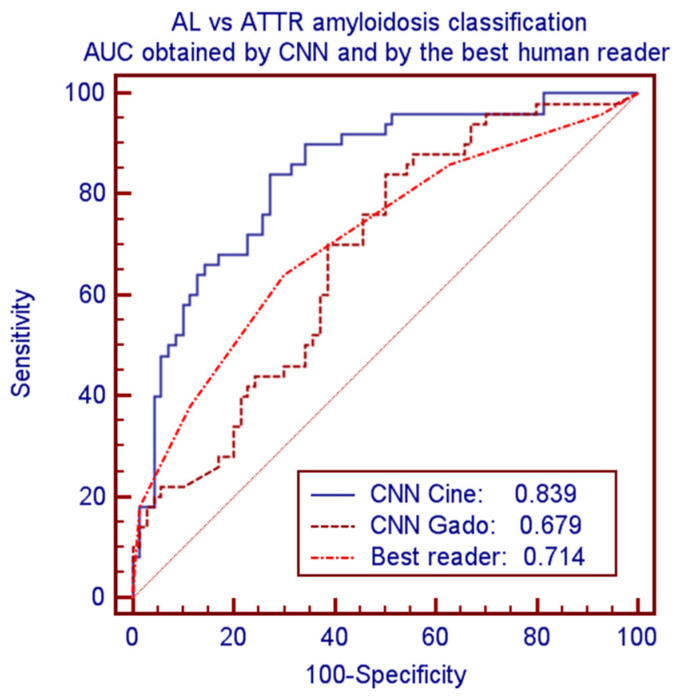
ROC curves and AUC for classification of AL vs. ATTR amyloidosis by CNN-cine, CNN-Gado and by the best human reader.

**Figure 5 biomedicines-11-00193-f005:**
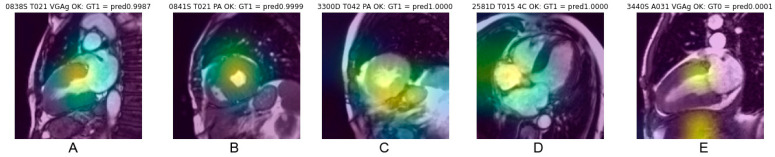
Example of saliency maps obtained in frames correctly classified, with heat area located in the cardiac region. Centering on the myocardial walls corresponds to the most consistent targeting (**A**,**B**). Targeting also frequently concerns a fairly large cardiac region covering the myocardium and the cardiac cavity (**C**). A topography favoring the cardiac cavities rather than the myocardium (**D**) or also targeting extracardiac areas (**E**) explain the relevance of the classification less well.

**Table 1 biomedicines-11-00193-t001:** Clinical and CMR characteristics of the study population.

	AL Amyloidosis	ATTR Amyloidosis	*p*
N patients	70	50	
Age (years)	72.1 ± 9.7	75.9 ± 9.4	0.034
Sex (M/F)	47/23 (67%)	47/4 (94%)	0.0004
Weight (kg)	68.7 ± 15.1	77.8 ± 13.1	0.0007
Height (cm)	170.1 ± 9.2	172.7 ± 9.4	0.14
BSA (m^2^)	1.81 ± 0.24	1.95 ± 0.19	0.0007
IVS (mm)	16.8 ± 3.0	19.7 ± 3.2	0.0001
LVMI (g/m^2^)	107.9 ± 31.0	125.4 ± 26.6	0.0017
LVDVI (ml/m^2^)	68.3 ± 23.7	74.9 ± 20.0	0.11
LVEF (%)	60.8 ± 10.5	56.9 ± 12.2	0.06
LA surface (cm^2^)	29.2 ± 5.9	31.1 ± 7.3	0.16
T1 (ms) (n = 48 vs. 42)	1146.7 ± 77.9	1143.3 ± 54.0	0.8
ECV (%) (n = 36 vs. 35)	50.7 ± 13.1	58.1 ± 13.1	0.019
T2 (ms) (n = 12 vs. 19)	51.5 ± 4.4	50.7 ± 2.3	0.49
N cine frames/patient	5.8 ± 1.8	5.9 ± 1.7	0.70
N LGE frames/patient	15.5 ± 4.8	16.0 ± 4.7	0.53
N patient with pericard	36 (51%)	13 (26%)	0.058
N patients with pleural	28 (40%)	8 (16%)	0.0073
N patients with both	16 (21%)	3 (6 %)	0.020

Characteristics of patients with AL and ATTR amyloidosis included in this study. Number of observation (integer) or average values ± standard deviation are listed. BSA: body surface area. IVS: interventricular septum thickness, LVMI: left ventricular mass index, LVDVI: left ventricular diastolic volume index, LVEF: left ventricular ejection fraction, LA: left atrial, ECV: extracellular volume. Between parenthesis is the percentage. Pericard is for pericardial effusion, pleural is for pleural effusion and both is for pericardial + pleural effusions.

**Table 2 biomedicines-11-00193-t002:** Performance of CNN classification between AL and ATTR cardiac amyloidosis.

	Accuracy	AUC
CNN-cine	0.750	0.839 [0.761–0.900]
CNN-LGE	0.611	0.679 [0.588–0.761] (0.0041)
Cine and LGE Average	0.708	0.821 [0.740–0.885] (0.42)
Cine and LGE Logistic regression	0.742	0.818 [0.738–0.883] (0.03)

Between brackets are 95% confidence interval for ROC AUC. Between parentheses are the significant level of statistical difference between CNN-cine and other results (assessed by Delong test for AUC comparisons).

**Table 3 biomedicines-11-00193-t003:** Accuracy and AUC of the ROC curve for classification of AL vs. ATTR cardiac amyloidosis for human readers vs. CNN-cine.

	Accuracy	ROC AUC
CNN-cine	0.750	0.839 [0.761–0.900]
Reader 1	0.633	0.622 [0.529–0.709] (0.0003)
Reader 2	0.617	0.644 [0.552–0.729] (0.0014)
Reader 3	0.675	0.714 [0.523–0.649] (0.0075)

Between brackets are 95% confidence interval for ROC AUC. Between parentheses are the significant level of statistical difference between CNN-cine and other results (assessed by Delong test for AUC comparisons).

## Data Availability

Database and code can be made available on reasonable request, after agreement of the Clinical Research Department of our hospital.

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
