# Peer review of "Deep Learning to Classify AL versus ATTR Cardiac Amyloidosis MR Images"

_biomedicines, 2023, doi:10.3390/biomedicines11010193_

Round 1

Reviewer 1 Report

The work done by the authors is very valuable from a scientific point of view. Illustrations and diagrams (for example, Figure 1. on the fifth page) significantly supplement the work. Statistics and analytical presentation of work results are at a high methodological level. The article will undoubtedly generate a lot of interest among experts and will certainly achieve a high citation rate.

Author Response

Decembre 18th 2022

Dear Editor,

Thank you for considering our article “Deep learning to classify AL versus ATTR cardiac amyloidosis MR images”, submitted to Biomedicines.

Please find the revised version of the article :

We would like to thank the reviewers for their commentary work and for their recommendations which we have taken into account for the revised version of the article.

We hope that these corrections will satisfy reviewers and the Editor and we remain at your disposal.

Sincerely

Ph Germain

Reviewer 2 Report

The paper is well written and the topic is of interest; the investigations well conducted, and appropriately decribed and discussed.

Author Response

(The authors gave the same response as above.)

Reviewer 3 Report

I read with great interest the work from Germain et al., describing the role of deep learning in differentiating AL from ATTR cardiac amyloidosis (CA).

The article is well written, the text is clear and supposedly the aim of the study is reached, which is to prove that to some extend deep learning is better than the human eye to differentiate two kinds of CA.

However, one may question the very basic concept of this trial:

-        First, it may not be as useful as expected, as a non-invasive algorithm including blood samples and bone scintigraphy already exists and is accurate. Therefore, unlike what the authors discuss, CMR has only little place in the diagnostic algorithm of CA (it is mostly used for differential diagnosis if both AL and ATTR have been ruled out, or for its prognostic value in CA). Moreover, a wide overlap exists among the various parameters, leading to two questions: do the physicians really step forward in clinical practice about of the etiology of CA when they perform a CMR, and if yes, on what criteria? And secondly, considering its low accuracy (confirmed by the AUC as low as 0.61, which is almost random), how relevant is the comparison between a physician eye and deep learning?

-        Second, I noticed that the two groups are not strictly comparable, as the ATTR form seems more evolved, based on the LVEF, ECV, the IVS and the mass. Therefore, my main concern would be that the deep learning would only differentiate the degree of evolution of the disease rather than its etiology. Why didn’t the authors perform a pairing on these criteria?

On the form:

-        Numbers in table 1 should be rounded to be relevant: IVS in mm, mass to g/m², height to cm, weight to kg or only one decimal point and so on…

-        Line 270 in the discussion: I would say “the diagnosis of CA could be easy in advanced forms of the disease”, as early forms are still underdiagnosed, and as it is the population studied in the trial

-        I would suggest removing the part about the weight in the discussion, as it blurs the message and is not relevant in clinical practice

Author Response

Decembre 18th 2022

Dear Editor,

Thank you for considering our article “Deep learning to classify AL versus ATTR cardiac amyloidosis MR images”, submitted to Biomedicines.

Please find the revised version of the article :

We would like to thank the reviewers for their commentary work and for their recommendations which we have taken into account for the revised version of the article.

The remarks listed below have been incorporated into the new manuscript

However, one may question the very basic concept of this trial:

-        First, it may not be as useful as expected, as a non-invasive algorithm including blood samples and bone scintigraphy already exists and is accurate. Therefore, unlike what the authors discuss, CMR has only little place in the diagnostic algorithm of CA (it is mostly used for differential diagnosis if both AL and ATTR have been ruled out, or for its prognostic value in CA). Moreover, a wide overlap exists among the various parameters, leading to two questions: do the physicians really step forward in clinical practice about of the etiology of CA when they perform a CMR, and if yes, on what criteria? And secondly, considering its low accuracy (confirmed by the AUC as low as 0.61, which is almost random), how relevant is the comparison between a physician eye and deep learning?

Yes, we agree that this work should in no way be considered as an alternative to the well-established diagnostic algorithm based on biological data and bone scintigraphy for cardiac amyloidosis. The aim of the work was only to position deep learning in relation to human visual abilities to discriminate between the two main varieties of amyloidosis.

The initial reason for which we undertook this work resulted from our amazement at the abilities of CNNs to discriminate pathologies indistinguishable by the naked eye, as reported in the literature and this singular capacity is the subject of the majority of the discussions in the article. Even if the performances obtained remain limited, in this work we show that the neural network performs approximately 10 points better (accuracy and AUC) than a human operator. As such, this work aims to highlight the advantages of deep learning compared to human diagnosis, without claiming to compete with the usual diagnostic algorithm.

Similarly, the use of MRI does not mean that it is the best diagnostic method to establish cardiac amyloidosis. The focus of the work was not clinical but methodological, focusing on the superiority of deep learning over the naked eye. In fact, MRI should only be considered here as the material, the substrate with which the demonstration could be made. 

-        Second, I noticed that the two groups are not strictly comparable, as the ATTR form seems more evolved, based on the LVEF, ECV, the IVS and the mass. Therefore, my main concern would be that the deep learning would only differentiate the degree of evolution of the disease rather than its etiology. Why didn’t the authors perform a pairing on these criteria?

Matching the two groups of amyloidosis would be very penalizing by drastically reducing the number of analyzable observations and this number is critical in deep learning which in principle uses several thousand observations (limitation that was pointed line 407 in the discussion). Moreover, the clear differences that exist between our two groups of AL and TTR amyloidosis (e.g. septal thickness or mass, pericardial effusion…) should not be considered as confounding factors but rather as useful pathological signatures (similarly reported in other articles [31-33]). As we point out in the discussion (line 411), these differences constitute discriminating features between the two types of amyloidosis, probably used by the neural network (but unfortunately, we do not know to what extent) and similarly used by the human operator.

-        Numbers in table 1 should be rounded to be relevant: IVS in mm, mass to g/m², height to cm, weight to kg or only one decimal point and so on…

Numbers in table 1 have been rounded with only one decimal place

-        Line 270 in the discussion: I would say “the diagnosis of CA could be easy in advanced forms of the disease”, as early forms are still underdiagnosed, and as it is the population studied in the trial

we have corrected the sentence (in line 273), according to the judicious suggestion of the reviewer.

-        I would suggest removing the part about the weight in the discussion, as it blurs the message and is not relevant in clinical practice

The sentence concerning the weight has been removed in accordance with the suggestion of the referee (line 282).

We hope that these corrections will satisfy reviewers and the Editor and we remain at your disposal.

Sincerely

Ph Germain

Round 2

Reviewer 3 Report

I would like to thank the authors for their answer, although change in the manuscript is minimal.

-        In my opinion, the design of the trial is flawed, and I have to disagree with their argument: as I previously stated, it is possible that the AI only diagnoses the morphological differences between ATTR and AL CA. For example, previous trials showed that they could differentiate ATTR and AL amyloidosis with higher accuracy than the AI presented here, based on the morphological differences and LGE (for example the QALE score https://doi.org/10.1016/j.jcmg.2013.08.015). Therefore, if the authors wanted to differentiate with AI both types of amyloidosis solely based on criteria that cannot be observed with the human eye, they would have – as I stated before – to perform pairing of these patients. This is even more true that, as the authors discussed, the “black box” of AI won’t let us know what analysis was performed.

-        The authors did not answer on what criteria they diagnosed ATTR or AL CA on CMR?